# From Recharge, to Groundwater, to Discharge Areas in Aquifer Systems in Quebec (Canada): Shaping of Microbial Diversity and Community Structure by Environmental Factors

**DOI:** 10.3390/genes14010001

**Published:** 2022-12-20

**Authors:** Karine Villeneuve, Michel Violette, Cassandre Sara Lazar

**Affiliations:** Department of Biological Sciences, University of Québec at Montréal, UQAM, C.P. 8888, Succ. Centre-Ville, Montréal, QC H3C 3P8, Canada

**Keywords:** groundwater, aquifer recharge, aquifer discharge, bacteria, archaea, temporal succession

## Abstract

Groundwater recharge and discharge rates and zones are important hydrogeological characteristics of aquifer systems, yet their impact on the formation of both subterranean and surface microbiomes remains largely unknown. In this study, we used 16S rRNA gene sequencing to characterize and compare the microbial community of seven different aquifers, including the recharge and discharge areas of each system. The connectivity between subsurface and surface microbiomes was evaluated at each site, and the temporal succession of groundwater microbial communities was further assessed at one of the sites. Bacterial and archaeal community composition varied between the different sites, reflecting different geological characteristics, with communities from unconsolidated aquifers being distinct from those of consolidated aquifers. Our results also revealed very little to no contribution of surface recharge microbial communities to groundwater communities as well as little to no contribution of groundwater microbial communities to surface discharge communities. Temporal succession suggests seasonal shifts in composition for both bacterial and archaeal communities. This study demonstrates the highly diverse communities of prokaryotes living in aquifer systems, including zones of groundwater recharge and discharge, and highlights the need for further temporal studies with higher resolution to better understand the connectivity between surface and subsurface microbiomes.

## 1. Introduction

The terrestrial subsurface holds the largest reservoir of liquid freshwater on Earth and about 2.5 billion people worldwide depend solely on groundwater as a source of freshwater [1]. As this number is expected to significantly increase in the following years, understanding the biotic and abiotic drivers of subsurface groundwater ecosystems is becoming crucial [2]. From a hydrogeochemical perspective, groundwater recharge (GWR) and discharge (GWD) rates and zones are considered important characteristics of aquifer systems [3,4,5]. Recharge refers to water entering a groundwater flow through direct infiltration of precipitation or infiltration through streambeds or reservoirs. Discharge refers to water exiting the groundwater flow into surface water bodies (e.g., spring, seep, stream, lake), different subsurface formations, or by pumping from a well. Through these two mechanisms, continuous dynamic interactions between surface and subsurface aquatic ecosystems occur within the water cycle [6]. However, the impact of GWR and GWD on the formation of both subterranean and surface microbiomes remains largely unknown.

Groundwater flow occurs in aquifers, which are defined as heterogeneous permeable geological formation capable of storing significant amounts of groundwater [7]. Aquifers exist in multiple forms and sizes and can be composed of unconsolidated sand and/or gravel, permeable consolidated deposits (such as sandstone and limestone), or consolidated less-permeable fractures rocks such as granitic and metamorphic rocks [7]. This heterogeneity of groundwater ecosystems offers both diverse and vast habitats for microorganisms, and it is likely that each environment supports distinct microbial communities well adapted to the local biotic and abiotic conditions [8].

Furthermore, certain geological characteristics of aquifers, such as surface connectivity, rock type, and porosity, can lead to important temporal variation in groundwater chemistry, as well as microbial community diversity and composition [9,10,11]. These variations are associated with strong seasonal weather events, such as high precipitation and melting snow, which can increase aquifer recharge. During such events, soil-derived microorganisms can be transported by water flowing into the subsurface ecosystem, where they can thrive and increase in abundance if conditions are suitable [1,10,12]. Additionally, through groundwater flow path, microorganisms may be transported from recharge to discharge areas. Thus, microbial communities from both recharge area and aquifers could be potential sources of microorganisms to discharge areas [5].

In this study, we compared the composition of groundwater bacterial and archaeal communities based on 16S rRNA amplicon sequence variants (ASVs) in seven different aquifer sites, including recharge and discharge areas. At one of the sites, the temporal succession of groundwater microbial communities was also assessed, thus allowing us to track the contribution of different sources to the aquifer’s prokaryote communities. Additionally, at each site, we estimated the potential contribution of soil, snow, and groundwater communities to surface water communities sampled in rivers, lakes, or streams. Our results show that microbial community composition differed in the studied aquifer ecosystems displaying distinct geological characteristics. Temporal succession highlighted shifts in groundwater microbial communities from summer to winter and spring. Spatial succession suggested a low contribution of surface to subsurface communities and a low contribution of groundwater to surface discharge communities.

## 2. Materials and Methods

### 2.1. Study Sites

Seven aquifer sites were sampled in the summers of 2018 and 2019 (Figure 1). The site close to the city of Vaudreuil (V; 45°28′49.2″ N, 74°15′37.2″ W) and the one in the city of Saint-Lazare (SL; 45°38′47.03″ N, 74°19′70.37″ W) were sampled in July 2018. In addition, V was sampled three other times throughout the 2018–2019 seasons. The aquifers at Covey Hill (CH; 45°0′27.41″ N, 73°49′7.51″ W), and two sites from the Kenauk national reserve (K1; 45°45′4.351″ N, 74°49′35.65″ W; and K2; 45°47′51.917″ N, 74°46′16.579″ W) were sampled in August 2018. Finally, two sites at Ascension close to the Rivière Rouge river (RR; 46°36′42.66″ N, 74°47′3.911″ W) and at Notre-Dame-des-Laus (ND; 46°0′59.148″ N, 75°34′37.667″ W) were sampled in July 2019.

The V and SL aquifers are unconfined and fed by surface water infiltration (personal communication, M. Larocque, UQAM). The V aquifer is located downstream of the Raquette river, and groundwater–river connections have previously been established (groundwater discharge) [13]. Groundwater flows through a clay layer of soil and is 2 m deep. The aquifer is thus poorly drained, favoring surface runoff rather than infiltration, and surface water recharge is lower than at SL. This aquifer is influenced by human activities such as farming and a nearby road. The SL aquifer is in sandy soils, with higher water drainage, and is 9 m deep. The SL aquifer also has accessible surface discharge areas in a forest slope, which then pour out into a river. The CH aquifer is composed of deformed and fractured sandstone [14]. Groundwater flows through fractures in the sandstone and is 5 m deep. Where the fractures converge with surface ground, groundwater discharges into small streams. Furthermore, the CH aquifer is hydrologically connected to a surface peat bog [15]. Both K aquifers are found in tonalitic gneiss under a thin layer of till. K1 is 30 m deep, and K2 is 60 m deep (personal communication, M. Larocque, UQAM). Both groundwaters flow into lakes (Poisson Blanc for K1, and Papineau for K2). The RR aquifer is mostly composed of sand with a smaller amount of clay than the ND aquifer. Both aquifers are located in a forested area [16]. RR is 18 m deep and ND is 1.75 m deep. The RR groundwater discharges in the Rivière Rouge river, whereas the ND groundwater seeps into a lake.

### 2.2. Sampling

At each site, aquifer recharge areas were sampled in the form of soil, apart from the CH site where water from the peat bog was collected (Table 1). The soil was collected with a sterilized spoon, and stored in 50-mL Falcon tubes on ice, until being returned to the lab where the soils were stored at −20 °C. In February 2019, we sampled snow at the V and SL sites. At each site, groundwater was collected by using a submersible pump (12V/24V Mini-Monsoon, Waterra, Mississauga, Canada). We waited for the physicochemical parameters to stabilize and discarded the initial groundwater flowing up as stagnant water. Temperature, pH, dissolved oxygen, and conductivity were measured in the field with a YSI multiparameter probe (model 10102030, Yellow Springs, OH, USA). For the V aquifer, a temporal study was conducted, and additional groundwater was collected in February, April, and May 2019 (Table 1). Groundwater was also collected in May 2019 from the SL aquifer, but the snow rendered the site inaccessible by car for sampling before this date.

River and lake samples were recovered from the shore. Groundwater discharge samples were collected at the Saint-Lazare site, at the bottom of the forest slope, where groundwater directly emerges to the surface. We sampled three areas, one below a tree root (d1), and two on the ground (d2a and d2b).

The CH peat bog water, the snow, the groundwater, and the lake and river water samples were collected in sterilized polypropylene bottles (Nalgene, Rochester, NY, USA), transported on ice, and stored at 4 °C until filtration in the lab, which was done the same day as sampling. Filtration was carried out by using 0.2-µm polyethersulfone filters (Sartorius, Germany). For filtration we used, 1 L of river/lake water, 4 L of groundwater, and 50 mL of groundwater discharge from the SL site. We also collected 1 L of snow which was thawed at room temperature in the lab before filtration. Filters were subsequently stored at −20 °C.

### 2.3. Geochemical Analyses

Water samples were collected in gas-free glass bottles after filtration on 0.45 µm polyethersulfone filters (Sarstedt ^®^, Numbrecht, Germany), to measure dissolved organic and inorganic carbon (DIC/DOC). Filtered water was also collected to measure ammonium. For DIC/DOC, samples were analyzed with an OI Analytical Aurora 1030W TOC Analyzer by using a persulfate oxidation method. For ammonium, samples were analyzed with a Flow Solution 3100 autosampler by using a chloramine reaction with salicylate to form indophenol blue dye (EPA Method 350.1). Both geochemical analyses were conducted at the Université du Québec à Montréal (UQAM)-GRIL analytical laboratory.

### 2.4. DNA Extraction, Illumina Sequencing and Sequence Analysis

DNA was extracted from the filters by using the DNeasy power water kit (Qiagen, Germany) according to the manufacturer’s instructions. DNA was extracted from 250 mg of soil by using the DNeasy power soil kit (Qiagen, Hilden, Germany). All extracted DNA samples were stored at −20 °C until further use.

Sequencing was carried out on the Center for Excellence in Research on Orphan Disease—Foundation Courtois (CERMO-FC) genomic platform at UQAM. Bacterial and archaeal 16S rRNA genes were amplified by using the polymerase UCP hiFidelity PCR kit (Qiagen, Germany) for a better sensitivity for low DNA concentrations. The V3-V4 region of the bacterial 16S rRNA gene was targeted by using the B341F (5′-CCTACGGGAGGCAGCAG-3′, [17])-B785R (5′GACTACCGGGGTATCTAATCC-3′, [18]) primer pair. The V3-V4-V5 region of the archaeal 16S rRNA gene was targeted by using the A340F (5′-CCCTACGGGCYCCASCAG-3′, [19])-A915R (5′-GTGCTCCCCCGCCAATTCCT-3′, [20]) primer pair. PCR amplification was performed under the following conditions: denaturation at 98 °C for 30 s, annealing for 30 s (57 °C for bacteria, and 58 °C for archaea), extension at 72 °C for 1 min, and final extension at 72 °C for 10 min. We used 33 cycles for the bacteria, and 40 cycles for the archaea. Sequencing was performed by using an Illumina MiSeq 2300 and the MiSeq reagent kit v.3 (600 cycles, Illumina, San Diego, CA, UQA). Negative controls for the PCR amplifications of both the bacteria and archaea were sequenced as well. All sequences were deposited on the National Center for Biotechnology Information platform (NCBI) under the BioProject ID PRJNA883426.

### 2.5. Sequence Analyses

The obtained sequences were analyzed by using mothur software v.1.44.3 [21]. Pair ends were merged, resulting in 575 bp sequences for the archaea, and 445 bp for the bacteria, and sequences were subsequently treated to detect and delete sequencing errors and chimeras. The dataset was then classified by using the SILVA database v.138.1 [22]. The classification of the archaeal 16S rRNA genes was further implemented with reference sequences from the Bathyarchaeaota [23] and the Woesearchaeota [24], as well as a personal database. Amplicon sequence variants (ASVs) were computed by using mothur. Rarefaction was carried out by using the median sequencing depth method [16,25], and we only kept samples with more than 1000 sequences (details can be found in Appendix A). Before running rarefaction analyses, we subtracted the ASVs that were sequenced in the PCR negative control from all samples.

### 2.6. Statistical Analyses

Shannon diversity indices (α-diversity) were calculated by using mothur. Indices were compared using a Kruskal–Wallis test with the dunnTest function of the FSA package, in R [26] v.4.1.2 (Appendix A). Community composition (ß-diversity) was observed with principal coordinate analysis (PCoA), run in mothur, by using the rarefied ASV table and a Bray–Curtis dissimilarity distance matrix. Analysis of molecular variance (AMOVA) was used on the distance matrix in mothur, to test whether the sample clustering observed on the PCoA plot was statistically significant.

To test whether community composition varied significantly depending on environment parameters, we ran permutational multivariate analyses (PERMANOVA) on the rarefied ASV tables in R by using the adonis function of the vegan package. Significantly different ASVs between sample groups were identified by using linear discriminant effect size (LEfSe) [27] analyses by using the online tool from the Huttenhower lab (https://huttenhower.sph.harvard.edu/lefse/, accessed on 12 January 2022).

We used distance-based redundancy analysis (db-RDA) to determine which variables had a significant impact on community composition in the groundwater samples. ASV tables were transformed and used to calculate a Bray–Curtis dissimilarity distance matrix. The explanatory variables included environmental parameters (dissolved organic carbon, ammonium, pH, temperature, and dissolved oxygen), as well as community composition of the bacteria when analyzing the archaeal community, and vice versa. Indeed, biotic factors such as other microbes present will also influence community composition. Community composition was represented by using the scores of the first two axes of a PCoA ordination [28]. Nonnormal variables were box-cox-transformed to approach a normal distribution. The db-RDA was applied to the distance matrix and the set of explanatory variables using the capscale function of the vegan package in R, and significance of explanatory variables was assessed with the ANOVA function in R with 200 permutations. The unique and shared contributions of each significant variable to community composition was determined by using variance partitioning with the varpart function of the vegan package in R. We used the fast expectation-maximization for microbial source tracking (FEAST) method [29] to estimate the source of microbial communities to the groundwater and surface discharge sites by using raw ASV tables without transformation.

## 3. Results

### 3.1. Groundwater Environmental Parameters

Regarding the samples collected in the summer, the groundwater from CH contained extremely high concentrations of dissolved organic carbon (DOC) and RR contained the least (Appendix A). Ammonium was below detection at V and SL, and highest at ND. pH was acidic at CH, V, and RR, and close to neutral at the other sites. Temperature was above 10 °C at CH, K1, K2, and RR, and below 8 °C at V and SL. The sites at V, K1, K2, RR, and ND were below 22% dissolved oxygen (DO), whereas SL and CH were above 82% DO. At V, over the four sampled time periods (summer to winter and spring), ammonium, pH, and DO increased, and temperature decreased.

### 3.2. Archaeal and Bacterial 16S rRNA Gene Taxonomy

The groundwater from V collected in July 2018 contained unclassified Woesearchaeales, candidatus (cand.), Nitrosotalea, unclassified (unc.) Thermoplasmata, and Woesearchaeota subgroups 3, 5a, and 24 (Figure 2a). All these taxa varied in relative abundance throughout the seasonal timepoints. The unc. Woesearchaeales dominated in July 2018 (16.3% of the total sequences), the unc. Thermoplasmata had the highest relative abundance in February 2019 (19.9%) and then decreased, and cand. Nitrosotalea was the dominating taxon in April and May 2019 (19.77 and 25.14%), increasing over time. The Woesearchaeaota subgroups 5a and 24 increased over time and were mainly present in April and May. The groundwater from SL collected in July 2018 contained cand. Nitrosotalea and Woesearchaeota subgroup 5a. The groundwater sample from CH collected in August 2018 contained Woesearchaeota subgroups 5a and 5b, Bathyarchaeota subgroups 6 and 18, the CG1-02-32-21 Micrarchaeota, and cand. Nitrosotalea. The groundwater from K1 collected in August 2018 contained *Nitrosoarchaeum*, *Methanobacterium*, and Woesearchaeota subgroup 5a. The groundwater from K2 collected in August 2018 contained Woesearchaeota subgroups 5a and 5b and cand. Nitrosotalea. The groundwater from RR collected in July 2019 contained cand. Nitrosotalea and Woesearchaeota subgroups 5a and 24. The groundwater from ND collected in July 2019 contained Bathyarchaeota subgroups 5bb and 18.

The groundwater discharge sample collected under the tree root at SL in July 2018 contained Woesearchaeota subgroups 8 and 20, and unc. Woesearchaeales (Figure 2a). Both discharge samples collected on the forest ground contained cand. Nitrosotalea, Woesearchaeota subgroup 24, group 1.1c Crenarchaeota, and Bathyarchaeota subgroup 18. The river at V collected in July 2018 contained Woesearchaeota subgroups 5a, 5b, 8, and 10, *Methanospirillum*, and unc. Woesearchaeales (Figure 2). In April 2019, the water contained Woesearchaeota subgroups 5a, 5b, *Methanosaeta*, and unc. Nitrososphaeraceae. In May 2019, the water contained Woesearchaeota subgroup 5a, *Methanosarcina*, *Methanosaeta*, *Methanoregula*, and Bathyarchaeota subgroup 6. The river at SL collected in July 2018 contained Woesearchaeota subgroups 5a, 5b, and 20, and unc. Woesearchaeales. In May 2019, the water contained Woesearchaeota subgroups 5a, 5b, and cand. Nitrosotalea. The stream at CH collected in August 2018 contained Woesearchaeota subgroups 5a and 5b, and *Methanobacterium*. The rivers at K1 and K2 collected in August 2018 contained Woesearchaeota subgroups 5a and 5b, and unc. Woesearchaeales. The river at RR collected in July 2019 contained Woesearchaeota subgroups 5a, 5b, and 8, *Methanoregula*, and Bathyarchaeota subgroup 6. The lake at ND collected in July 2019 contained *Methanosaeta*, *Methanoregula*, *Methanobacterium*, *Methanosarcina*, and Woesearchaeota subgroup 5b.

The groundwater from V collected in July 2018 contained unc. Bacteriovoracaceae, unc. Oxalobacteraceae, and *Nitrospira* (Figure 2b). *Pseudoarthrobacter*, *Sphingomonas*, *Bacillus*, and unc. Micropepsaceae were mostly detected in February 2019, while unc. Oxalobacteraceae were the highest taxon in terms of relative abundance in April and May 2019 (59.02 and 38.53% of the total sequences). The groundwater from SL collected in July 2018 contained *Sphingomonas* and *Methylobacterium*-*Methylobrum*. The groundwater sample from CH collected in August 2018 contained unc. Acidimicrobiia, cvE6 Chlamydiales, unc. Acetobacteraceae, unc. Vermiphilaceae, and *Sideroxydans*. The groundwater from K1 collected in August 2018 contained Bivii28 Williamwhitmaniaceae, hgcl_clade Sporichtyaceae, *Desulfovibrio*, *Curvibacter*, *Nitrospira*, and *Geothrix*. The groundwater from K2 collected in August 2018 contained Citrifermentans, Sideroxydans, cvE6 Chlamydiales, and Lineage_IV Elusimicrobia. The groundwater from RR collected in July 2019 contained *Alkanindiges*, *Rhodoferax*, *Duganella*, unc. Oxalobacteraceae, *Polynucleobacter*, *Sulfuricurvum*, *Undibacterium*, and *Limnohabitans*. The groundwater from ND collected in July 2019 contained unc. Dehalococcoidia, unc. Sva0485, *Acetobacterium*, cand. Omnitrophus, *Geobacter*, and *Desulfovibrio*.

The groundwater discharge sample collected under the tree root at SL in July 2018 was dominated by unc. Xanthobacteraceae, and IMC26256 Acidimicrobiia (Figure 2b). Both discharge samples collected on the forest ground contained *Nitrospira*, unc. Burkholderiales, MND1 Nitrosomonadaceae, and IMC26256 Acidimicrobiia. The river at V collected in July 2018 was composed of *Limnohabitans*, the hgcl_clade Sporichtyaceae, unc. Sporichtyaceae, *Sphingorhabdus*, *Pseudarcicella*, *Polynucleobacter*, *Sediminibacterium*, and *Rhodoluna*. In April 2019, the water contained unc. Oxalobacteraceae, *Polaromonas*, *Flavobacterium*, *Rhodoferax*, and TM7 Saccharimonadales. In May 2019, the water contained *Flavobacterium*, *Bacillus*, *Flavobacterium*, Pseudoarthrobacter, unc. Comamonadaceae, and *Polaromonas*. The river at SL collected in July 2018 contained *Rhodoferax*, unc. Comamonadaceae, *Limnohabitans*, and *Flavobacterium*. In May 2019, the water contained *Flavobacterium*, *Rhodoferax*, unc. Comamonadaceae, *Polaromonas*, and *Limnohabitans.* The stream at CH collected in August 2018 contained the hgcl_clade Sporichtyaceae, *Polynucleobacter*, *Undibacterium*, and *Methylocysis*. The rivers at K1 and K2 collected in August 2018 contained *Flavobacterium*, *Rhodoferax*, and unc. Comamonadaceae. The river at RR collected in July 2019 contained *Polynucleobacter*, unc. Sporichtyaceae, the hgcl_clade, *Limnohabitans, Sediminibacterium*, and cand. Planktoluna. The lake at NDDL collected in July 2019 contained CL500-29 Illumatobacteraceae, the hgcl_clade Sporichtyaceae, unc. Sporichtyaceae, *Limnohabitans*, and unc. Beijerinckiaceae.

### 3.3. Alpha-Diversity Indices and Comparison between Habitats

For the Archaea, diversity indices ranged between 4.81 and 5.7 in the soil samples, 2.73 and 7.17 in the groundwater samples, and 5.95 and 8.53 in the surface samples (Appendix A). For the bacteria, diversity indices ranged between 3.04 and 8.46 in the soil samples, 3.66 and 7.47 in the groundwater samples, and 5.36 and 8.28 in the surface samples (Appendix A).

We compared diversity indices between aquatic habitat types (groundwater, groundwater discharge area, and lake/river/stream surface water). For the archaeal communities, the only significant difference was found between the groundwater and lake/river/stream surface samples (*p* = 0.00027), with the groundwater samples being less diverse than the surface samples. For the bacterial communities, none of the comparisons were significant.

### 3.4. Beta-Diversity Community Composition in Aquatic Habitats

Ordination of the archaeal communities by using PCoA highlighted three clusters (CA1, CA2, CA3) distinguishing groundwater and groundwater discharge samples from SL (CA1), groundwater samples from V (CA2), and surface river samples from V, SL, CH, and K (CA3; Figure 3a). Ordination of the bacterial communities highlighted three clusters as well (CB1, CB2, CB3) distinguishing groundwater discharge samples and the July 2018 groundwater sample from SL (CB1), groundwater samples from V (CB2), and surface samples from V, SL, CH, and K (CB3; Figure 3b). AMOVA tests supported significant differences between the three clusters for both the archaeal and bacterial PCoA analyses (Appendix A).

By using LEfSe, for the archaea, we explained the difference between these clusters with the Woesearchaeota (subgroups 5a, 5b, 7, 8, 10, 11 and 20), and methanogenic genera (*Methanoregula*, *Methanobacterium*, *Methanosarcina*, *Methanosaeta*, *Methanospirillum*, and *Methanocella*) being significantly higher in cluster CA3, which contains the surface water communities (Appendix A). For the bacteria, *Rhodoferax*, *Limnohabitans*, *Sediminibacterium*, *Sphingorhabdus*, and *Bdellovibrio* were significantly higher in cluster CB3 (Appendix A).

PERMANOVA testing showed that when analyzing community composition from the V and SL aquifers, habitat type (groundwater, groundwater discharge, and surface river water) explained 29.9% of the archaeal variance, and 23.4% of the bacterial variance, whereas site (V or SL) explained 12.7% of the archaeal variance, and 8.9% of the bacterial variance (Table 2).

### 3.5. Microbial Community Correlation with Environmental Variables

Archaeal and bacterial communities were analyzed in the groundwater of seven different aquifer systems in Quebec (Canada), during the summer season (July/August). For both domains, the communities from the Vaudreuil (V) site were distinct from the communities at Saint-Lazare (SL) site, and both groups were distinct from the communities at Covey Hill (CH), both sites from the Kenauk natural reserve (K1 and K2), and both Laurentians sites—Rivière Rouge (RR) and Notre-Dame-des-Laus (ND)—as determined by the db-RDA ordination (Figure 4).

For the archaea, NH_4_ and temperature were significantly correlated with community composition (Appendix A), both explaining 6.41% of the archaeal variance (Appendix A) and seemed to best be associated with the CH/K/RR/ND aquifer cluster. Although it was not a significant correlation, the db-RDA suggested that bacterial community composition explains archaeal variance in the V cluster. For the bacteria, DOC, pH, NH_4_ and temperature were significantly correlated with community composition (Appendix A) explaining 8.64% of the bacterial variance in total (Appendix A). pH seemed to best be associated with the SL cluster, whereas DOC, NH_4_ and temperature were best associated with the CH/K/RR/ND aquifer cluster.

The LEfSe analysis shows that only the Bathy_6 archaeal taxon was significantly higher in the CH/K/RR/ND aquifer cluster compared to the other clusters, while unc. Thermoplasmata, cand. Aenigmarchaeum and cand. Iainarchaeum, as well as Woesearchaeota subgroups 3, 4, 21, and 22b were significantly higher in the V cluster (Figure 5a). For the bacteria, in the CH/K/RR/ND cluster, *Gaillonella*, unc. Desulfobacterota, unc. Zixibacteria, and unc. Nitrospirota were significantly higher, whereas unc. Gemmatimonadaceae, *Methylobacterium*/*Methylobrum*, unc. Nitrosomonadaceae, cand. Nitrotoga, subgroup 12 Acidobacteriota, and unc. Vicinamibacterales were significantly higher in the SL cluster (Figure 5b). Unc. Bacteriovoraceae, BSV26 Kryptoniales, TRA3_20 Burkholderiales, bacteria25 Myxococcota, Lineage IIc Elusimicrobiota, cand. Ovatusbacter, and *Peredibacter* were significantly higher in the V cluster.

### 3.6. Microbial Source Tracking

We tracked the source of the microbial communities in the groundwater of the V site during four time points: July 2018 (only for the bacteria) and February, April, and May 2019. For the July 2018 time point, we used the July 2018 soil and surface river samples as potential sources. For the other time points, we used the groundwater community from the previous time point(s), the February 2019 snow sample, the May 2019 soil sample for the May 2019 time point, and the river water samples as potential sources given that a connection between groundwater and river has been evidenced in previous studies at the V site [13]. In July 2018, most of the bacterial sources were unknown, with a 0.1% contribution of the soil community, and 0.018% from the river water (Figure 6b). In February 2019, a majority of the identified archaeal and bacterial communities originated from the July 2018 groundwater (10.3 and 23.9%, Figure 6a,b). The rest was contributed by the surface snow and the July 2018 soil.

In April, right after the spring snowmelt, most of the archaeal and bacterial community came from the July 2018 groundwater, February 2019 groundwater, and April 2019 river water (Figure 6). Finally in May, the microbial community came from the July 2018 groundwater, the February 2019 groundwater, a majority from the April 2019 groundwater, the April 2019 river water, and the May 2019 river water.

We also tracked the source of the microbial communities in the surface water samples (river, lake, or stream) at each site, using soil recharge (peat bog for the CH site), snow and groundwater communities as potential sources. For the river water at V and SL, we used the river communities from the previous time points as well (July 2018 for April 2019, and July 2018 and April 2019 for May 2019). Finally, we also used the groundwater discharge samples from SL as a potential community source to the river community in July 2018. For the bacteria, we also tracked the source in the groundwater discharge samples from SL by using soil and groundwater as potential sources. Because the K2 site surface recharge soils did not contain enough archaeal and bacterial sequences, we could not track the source of communities in the K2 river water.

In the summer of 2018, when using the surface soils and groundwater as potential sources for the surface river/lake/stream communities, the overall estimated sources are less than 10% for both archaea and bacteria (Figure 7). In July 2018, at the V site, the source of the bacterial community was mostly unknown, with a negligible contribution from the surface soil community (0.03%; Figure 7b). At the SL site, most of the estimated source came from the forest groundwater discharge water, the groundwater, and the soil for the bacteria (0.027%). At the CH site, much of the stream community originated from the surface peat bog (3.15 and 8.69%), and some from the groundwater. At the K1 river, the community came from the surface soil (3.14 and 2.87%), and the July groundwater. In July 2019, at the RR site, 1.81% of the bacterial community came from the groundwater. At the ND site, the vast majority of both the archaeal and bacterial communities were of unknown origin.

For the V site, in April 2019, the biggest source of the river water archaeal community came from the previous river time point sample (July 2018, 4.87%) with some contributions from the April groundwater and the February snow communities (Figure 7a). In May, 16.99% of the archaeal community originated from the previous river water communities, and from the groundwater and snow communities. For the bacteria, in April, the community originated from groundwater communities, with a contribution from the surface snow and soil communities (Figure 7b). In May, the community originated in the majority from the July (0.15%) and April (9.95%) river water, with a contribution from the surface snow and soil communities, and the groundwater community. At the SL site, the biggest source of the bacterial community for all three forest groundwater discharge water samples, was the groundwater (Figure 7b). In May 2019 at SL, the biggest part of the microbial river water communities came from the previous river time point communities (5.23% for the archaea, and 3.13% for the bacteria), with some contribution from the groundwater, and the forest discharge water (Figure 7).

## 4. Discussion

### 4.1. Microbial Communities Differ in Distinct Groundwater Ecosystems and Correlation with Environmental Parameters

The db-RDA showed that the groundwater microbial communities from the V and SL aquifer systems were dissimilar to all other aquifer sites, while communities at Covey Hill (CH), both sites from the Kenauk natural reserve (K1 and K2), and both Laurentians sites, Rivière Rouge (RR) and Notre-Dame-des-Laus (ND), were clustered in one group and hence shown to be similar in structure.

For the archaea, NH_4_ and temperature were correlated with community composition in the CH/K/RR/ND cluster, and the LEfSe analysis indicated that the Bathy_6 archaeal taxon was significantly higher in this cluster. This lineage is still uncultured, but genomic-based studies suggest it is a generalist archaeon adapted to both planktonic and sediment habitats [23,30]. Metagenomic reconstructions of this archaeon reveals it is a heterotroph able to degrade extracellular plant-derived saccharides, and it possesses a methyl-glyoxylate pathway typically activated when slow-growing cells are exposed to an increasing supply of sugar phosphates [23,30]. There is also evidence of dissimilatory nitrite reduction to ammonium (identification of genes nirB and nrfD) [30]. The aquifers of this cluster are either fractured or clay/sand unconsolidated systems, all recharged by forest soils or a peat bog for the CH aquifer, which are all heavily laden with plant-derived saccharides. The DO was low in most groundwater samples, which could explain the correlation with ammonium, or nitrite reduction with nitrite as a potential electron acceptor.

Although it was not a significant correlation, we observed a potential impact of the bacterial community on the archaeal community composition in the V cluster, where unc. Thermoplasmata, cand. Aenigmarchaeum and cand. Iainarchaeum, as well as Woesearchaeota subgroups 3, 4, 21 and 22b were significantly higher (LEfSe analysis). All these taxa also belong to uncultured lineages; however, metagenomes were reconstructed for most of these. The cand. Aenigmarchaeum and Iainarchaeum are conjectured to have limited catabolic capacities and to have symbiotic associations with bacterial hosts [31,32]. Some Thermoplasmatales have metabolic pathways allowing them to survive in contaminated waters [33]. Finally, the Woesearchaeota are widespread, with potential symbiotic and/or fermentation-based lifestyles [24]. The V aquifer is located in a farmland and is impacted by input from a nearby road. Therefore, the archaea in this site would be adapted to the input of surface contaminated water. Furthermore, in July 2018, the average precipitation was low (10 mm), and the average temperature was 25 °C. Thus, little precipitation and high temperatures would lead to a very low input of surface recharge soils to the groundwater, supporting archaeal limited metabolic capacities and symbiotic interactions with other organisms.

For the bacteria, the db-RDA showed that DOC, NH_4_, and temperature were significantly correlated with community variance, associated with the CH/K/RR/ND aquifer cluster, in which iron-oxidizing *Gaillonella*, sulfate-reducing Desulfobacterota [34], Zixibacteria, and Nitrospirota were significantly higher (LEfSe analysis). The higher temperatures (compared to the V and SL clusters) could explain the prevalence of the *Gaillonella* in this cluster, whereas the higher NH_4_ could allow nitrification mediated by Nitrospirota to occur [35]. The higher DOC could also explain the dominance of these heterotrophic bacteria [36]. pH was also significantly correlated with community variance, associated with the SL cluster, in which unc. Gemmatimonadaceae, *Methylobacterium*/*Methylobrum*, unc. Nitrosomonadaceae, cand. Nitrotoga, subgroup 12 Acidobacteriota, and unc. Vicinamibacterales were significantly higher (LEfSe analysis). Gemmatimonadota are found in soils and freshwater habitats, and are surmised to be adapted to dry soils, like the SL sandy soils (which contained a high relative abundance of Gemmatimonadota), as well as to prefer neutral pH over acidic pH, like the pH measured in the SL groundwater [37]. The neutral pH might also explain the significant presence of unc. Vicinamibacterales, which contain some taxa that are neutrophilic chemoheterotrophs [38]. The presence of nitrite-oxidizers (cand. Nitrotoga, [39]) is probably associated with the high relative abundance of the archaeal ammonia-oxidizing cand. Nitrosotalea. Finally, *Methylobacterium*/*Methylobrum* use C_1_ compounds (methanol, methylamine, formaldehyde) as the sole carbon source [40]. Methylamine is formed during biological degradation in terrestrial environments and emitted from natural vegetation [41,42]. Methylated amines can also be used as a nitrogen source for microbes [43]. Given that a coniferous forest is at the surface of the sandy SL aquifer, it is most likely that the groundwater receives C_1_ compounds released from the forest soils.

Most of the bacterial taxa found to be significantly higher in the V cluster are known predatory bacteria (Bacteriovoraceae, Myxococcota, and *Peredibacter*) [44,45], whereas cand. Ovatusbacter, which was also significantly higher in the V cluster, is an amoeboid symbiont [46]. These bacteria typically invade specific prey, although little is yet known about the predatory functioning of these organisms [47]. Given the probable oligotrophic conditions in the V groundwater in July, it is likely that preying on other microbes allows these taxa to gain carbon, nitrogen, or energy sources in oligotrophic environments [48]. The prevalence of predatory bacteria, as well as symbiotic archaea and bacteria, clearly demonstrates a flexibility of the microbial populations in response to their surrounding abiotic conditions.

Overall, these results show that the groundwater prokaryotic community composition was dissimilar between sites, depending on their aquifer geochemical, but also geological composition. We also observed that surface conditions, or rather the type of recharge soil or area (sandy, clay, farmland or forest, peat bog), shaped the subsurface microbial communities, probably by the nutrients or the microbial populations they provide [5,49]. Moreover, even though this data was not measured in this study, it is likely that the aquifer geology defined microbial community structure by controlling the intensity and speed of water flowing from the surface. Water will flow slowly in clay aquifers (V), compared to sandy or fractured aquifers (SL, CH and K). Water retention could explain the prevalence of predatory bacteria, and archaea with limited metabolic capabilities only detected at V. Previous studies have shown that environmental parameters explain a low fraction of microbial variance in groundwater habitats, and subsurface environments in general [5,16]. It has been suggested that biotic interactions (within organisms of the same domain, or between domains, symbiotic, predatory or competitive interactions) might explain some of the unaccounted variance, as well as dispersion [50]. Biotic interactions and dispersion were not measured in this study, but further work could help elucidate the participation of these factors on microbial variance.

### 4.2. Temporal Microbial Succession at the Vaudreuil Aquifer Site

We collected groundwater samples at four different timepoints at the V aquifer site: July 2018, and February, April, and May 2019. Comparing physico- and geochemical parameters from February to April 201 we observed that input of water in the subsurface from snow melting in the surface recharge areas led to an increase in pH, DOC, DO, and NH_4_, and a decrease in temperature. In November 2018, the average surface temperature was 0 °C, −5 °C in December 2018, −10 °C in January 2019, and −8 °C in February 2019, evidence that the surface soil was completely frozen three months prior to the groundwater sampling. This means that there was no input of surface water, nutrients, or microbial communities to the groundwater communities during that time. It is also highly likely that the water table decreased, although we did not measure this [51]. Unc. Thermoplasmata, *Pseudoarthrobacter*, *Sphingomas*, *Bacillus*, and unc. Micropepsaceae dominated the February 2019 samples. Genome reconstruction of *Thermoplasma acidophilum* has shown that this archaeon is adapted to scavenging nutrients from the decomposition of organisms [52]. Moreover, genome predictions have shown the presence of genes involved in osmotic pressure regulation [53], and the presence of genes coding a trehalose [54] that can protect cells from desiccation or dehydration [55]. Some strains of *Pseudoarthrobacter* are psychotropic and can grow in nutrient-poor environments [56,57]. *Sphingomas alaskensis* is an ultramicrobacterium, which is a strategy to cope with oligotrophic environments [58]. *Sphingomonas oligoaromativorans* can also grow in nutrient-poor habitats by degrading aromatic compounds such as benzoic, ferulic, or p-coumaric acid [59], all of which are known to be plant-derived molecules present in groundwater as a result of seeping from surface soils [60]. *Bacillus* can grow in oligotrophic aquatic ecosystems potentially thanks to physiological adaptive responses to low phosphorus concentrations [61]. Finally, *Micropepsis pineolensis* was isolated from an oligotrophic peat soil, is mildly acidophilic and is a fermenter [62]. Therefore, we observed a drastic change in the relative abundance of archaeal and bacterial taxa in the February groundwater, as the habitat probably became extremely oligotrophic. The microbial taxa detected in June 2018 were likely replaced by organisms able to withstand poor nutrient conditions by either scavenging for leftover recalcitrant organic matter or derived from dead cells, or by decreasing their cell size; the organisms are also able to endure dehydration.

In March 2019, the surface average temperature was −3 °C, 5 °C in April, and 12 °C in May. This means that by our sampling in April 2019, the snow had started to melt and to seep into the subsurface, as evidenced by a decrease in groundwater temperature from February to April, and an increase in pH, NH_4_, DOC, and DO. In the spring of 2019, cand. Nitrosotalea unc. Oxalobacteraceae dominated the groundwater samples. Cand. Nitrosotalea is an acidophilic ammonia-oxidizer [63], the relative abundance of which probably increased with the increase of NH_4_ in the spring, brought by the surface water. However, we cannot explain its presence in a neutrophilic environment. The pH in the February groundwater was 5.15, and contained sequences affiliated with the cand. Nitrosotalea, which could have acclimatized to the more neutral pH. Members of the Oxalobacteraceae are diverse, including nitrogen-fixing organisms. Bacteria belonging to this family were shown to be dominant during the initial phases of aquifer sediment colonization, during the month of May [64]. Furthermore, Oxalobacteraceae sequences were abundant in groundwater samples connected to agricultural fields, with widespread application of organic and inorganic fertilizers, similar to the V site [65]. The authors suggest that these bacteria are responsible for the reduction of nitrate seeping from the surface and are stimulated by the input of nitrate-rich waters. Finally, the use of deicing salt on roads has been shown to increase manganese transport to groundwater [66]. Bacteria of the Oxalobacteraceae that can oxidize manganese [67], could have been stimulated by the percolation of water impacted by the road close to the aquifer system in V. Therefore, we observed severe community shifts from summer to winter, and then spring, showing that surface conditions, or seasons, had a profound effect on microbial community composition.

### 4.3. Source of Groundwater Microbial Communities at the Vaudreuil Aquifer Site

We tracked the source of the microbial communities in the groundwater of the V site during the four analyzed time points: July 2018 (only for the bacteria) and February, April, and May 2019 by using microbial source tracking. In July 2018, most of the bacterial sources were unknown, with low contributions of the soil and river water communities. As mentioned above, in July 2018, the average precipitation was low, the average temperature was 25 °C, and the river water level was low at the time of sampling. Thus, the seeping of water from the surface to the subsurface was likely extremely limited, supporting our previous assumption of a very low input of surface areas to the groundwater.

In February 2019, a majority of the identified archaeal and bacterial communities originated from the July 2018 groundwater. As mentioned in the previous paragraph, the soil was and had been totally frozen for a few months when we sampled in February, explaining the almost unique contribution of the previous groundwater timepoint to the microbial community. The unc. Thermoplasmata that dominated the archaeal community, and the *Pseudoarthrobacter*, *Sphingomonas*, and *Bacillus* were also present in the July 2018 sample, but to a lesser extent (6.27, 0.07, 0.6, and 0.01%). This confirms that the oligotrophic setting characterizing the winter season in the groundwater, stimulated and promoted the growth of these taxa able to cope with the strict environmental conditions.

In April 2019, right after the spring snowmelt, and in May 2019, the archaeal and bacterial communities came into the majority from the previous groundwater timepoint communities. This shows that in both April and May, the contribution from the surface recharge soil communities, or even the surface snow communities, was negligeable compared to the involvement of the organisms already present in the groundwater at previous time points. This is also evidence of the flexibility of microbial communities facing fluctuating environmental settings. Based on our observations, it is likely that populations that belong to rare taxa at some time points (or seasons) are stimulated and promoted when conditions change [1,68,69,70,71]. The connection between the groundwater and river water is also supported here by an input of the river communities to the groundwater communities. The Woesearchaeota subgroup 5a dominated both the April and May river samples (developed in the next paragraph). The unc. Oxalobacteraceae dominated the April river sample, along with *Polaromonas* and *Rhodoferax*, whereas *Flavobacterium*, *Bacillus*, and *Rhodoferax* dominated the May river sample. This could in part explain the increase in Woesearchaeota 5a, *Polaromonas*, and *Rhodoferax* in the April groundwater; and of *Flavobacterium* in the May groundwater for the bacteria.

Our results show that there seems to be little to no input of surface communities in the groundwater. Alternatively, it also likely that the input from surface soils is high but given the drastic change in environmental conditions (little to no light, oligotrophic conditions, colder temperature), not many microbes survive this relocation. Added to the observations made in the previous paragraphs, the surrounding abiotic conditions, and possibly the biotic interactions between organisms seem to have a bigger impact on community composition than an input of new microorganisms from the surface.

### 4.4. Changes in the Microbial Community from Subsurface to the Surface Discharge Aquatic Ecosystems

Permutational multivariate analyses of variance (PERMANOVA) showed that habitat type (groundwater, groundwater discharge, and surface river water) explained the majority of archaeal and bacterial variance. Furthermore, diversity indices were significantly lower in groundwater compared to lake/river/stream surface samples for the archaeal communities. These results were not surprising not only given the pronounced environmental differences between surface and subsurface conditions, but also the more extreme conditions found in the subsurface ecosystems. Ordination of the archaeal and bacterial communities showed that communities from the surface river samples from V, SL, CH, and K were similar in terms of composition and dissimilar to the groundwater communities; and that the water microbial communities directly emerging in the forest surface at SL were similar in composition to the groundwater samples from SL (PCoA plot).

Woesearchaeota and methanogenic genera for the Archaea, and *Rhodoferax*, *Limnohabitans*, *Sediminibacterium*, *Sphingorhabdus*, and *Bdellovibrio* for the bacteria, were significantly higher in the surface river samples. Although methanogenic archaea are commonly viewed as strict anaerobes, there is increasing evidence that they are present in oxygenated waters in freshwater habitats (rivers or lakes) [72,73,74]. Methanogenesis in these ecosystems appear to be driven by biotic interactions with algae communities, or photoautotrophs, which provide acetate or H_2_, common substrates for methanogens. Woesearchaeota are widespread archaea and are a dominant group in surface aquatic systems [75,76]. Genome reconstructions points to metabolic deficiencies and symbiotic and/or fermentative lifestyles [77]. The presence of specific subgroups in our samples remains unexplained, as information on the distribution and physiology of these uncultured lineages still lack understanding. *Rhodoferax*, *Limnohabitans*, *Sediminibacterium*, and *Sphingorhabdus* are chemoheterotrophic bacteria typically found in surface aquatic freshwater habitats [78,79,80,81,82,83,84,85]. *Bdellovibrio* have been shown to be important predators in lake surface waters, probably mediating biotic interactions and microbial food webs [86].

### 4.5. Source of Microbial Communities in the Surface Discharge Aquatic Ecosystems

We tracked the source of the microbial communities in the surface water samples (river, lake, or stream) at each site, using soil recharge (peat bog for the CH site), snow, and groundwater communities as potential sources. In the summer of 2018, our results indicated that most of the surface aquatic communities were not sustained by subsurface microbial populations. At both rock-fractured sites, CH and K1, there was a larger impact of surface recharge area communities to the surface discharge habitats, evidence that water flowing faster compared to unconsolidated aquifers, brings a higher number of cells from the surface, to the subsurface and back to the surface [5]. The peat bog at CH was dominated by Woesearchaeota subgroups 5a and 5b, methanogenic genera *Methanobacterium*, *Methanoregula*, and the CG1-02-32-21 Micrarchaeota, and *Polynucleobacter* and *Undibacterium*, which are all typical peat bog and freshwater archaeal [73,80,81,87,88] and bacterial [89,90] taxa. Some strains of the *Polynucleobacter* genus are obligate endosymbionts [91], whereas some strains of the *Undibacterium* genus are endophytes [92]. The methanogens and *Polynucleobacter* were found at a relative abundance lower than 0.5% in the groundwater, whereas *Undibacterium* was not detected. This suggests that they were stimulated by favorable conditions once they reached the surface water environment, such as specific nutrients or hosts. The soil at K1 was also dominated by Woesearchaeota subgroups 5a and 5b, and the CG1-02-32-21 Micrarchaeota, as well as unc. Gailellales, Xanthobacteraceae, Gemmatimmonadales, and Acidimicrobiia. The same archaea dominated the river water community and the groundwater (except the Micrarchaeota which was less than 0.5%). However, only unc. Comamonadaceae and Burkholderiales were found at the same relative abundances in both the surface soil and the river water. Although we did not sample the surface soil near the river (it was sampled in the forest), we cannot exclude that the soil community did not follow a surface–subsurface–surface route but was introduced directly from the banks of the river.

Given the interconnection between groundwater and river at the V site, it is difficult to estimate whether the groundwater communities feed the river communities, or vice versa. This was very likely dependent on the season, because for both the archaea and the bacteria, the groundwater communities were more a source to the river communities in April, during the snowmelt season, than in May when the soil is unfrozen, and the snow has melted. The February 2019 snow archaeal communities were dominated by the Bathyarchaeota subgroup 6, the Woesearchaeota subgroup 5a, and the methanogenic genus Methanosaeta, which were also the dominant taxa in the April and May river water communities, but not the July 2018 communities, except *Methanosaeta*. The bacterial snow community was dominated by *Acidiphilium* and *Actinomycetospora*. Both these taxa were not detected in the river water in April and May, and it is unclear which group of bacterial taxa was involved in shaping the river communities. The same observation can be made for the involvement of the soil bacterial communities. Both *Acidiphilium* [93] and *Actinomycetospora* [94] are found in acidic habitats, comparable to the acidic pH in snow [95]. These genera do not seem to be found in neutral pH habitats, whereas the archaeal lineages that were detected in the snow are much more widespread and can be found in cold, warm, low, or neutral pH habitats [96,97,98,99,100]. Therefore, the archaea inhabiting the snow have the potential to acclimatize to fluctuating abiotic parameters, which is not the case of the bacteria. However, as mentioned for the K1 site, we cannot untangle whether the microbial snow community’s contribution to the surface river communities went from the surface recharge soils, to the subsurface and back to the surface, or if they originate from the snow above the frozen river that would have melted directly into the water in the spring.

At the SL site, the biggest source of the bacterial community for all three forest groundwater discharge water samples originated from the groundwater, and ordination analysis showed that the July 2018 groundwater and forest discharge community compositions were similar. This suggests that at the direct point of emergence at the surface, groundwater bacterial communities were still relatively similar in structure because they probably did not have time to acclimatize or adapt to the changing conditions.

In May 2019 at SL, the biggest part of the microbial river water communities came from the previous river time point communities Although it has been shown that river microbial communities shift rapidly due to water transit [101], our observations suggest that some of the community lasts over time. Because we always sampled the river water at the same location, this might be explained by similar environmental conditions that select the same taxa [102]. In addition, some of the cells could be attached to the riverbed or associated with a host (prokaryotic or eukaryotic). Overall, our results show that a minor proportion of the surface river water is fed by groundwater community, for both the archaea and the bacteria. This is most likely due to the drastic environmental change from subsurface to surface habitats.

## 5. Conclusions

We show in this study that aquifer site, and thus local geological features controlled both archaeal and bacterial community diversity and composition, with communities from unconsolidated aquifers being distinct from those of consolidated aquifers. This likely highlights an influence of subsurface water speed flow and intensity on groundwater communities. Both archaeal and bacterial communities were significantly correlated with NH_4_ and temperature, but geochemical parameters explained less than 10% of microbial variance. This suggests that other biotic parameters, such as species interactions, competition, predation, or symbiosis still need to be determined to help explain the observed community patterns. Surface conditions (seasons or land type of recharge areas) seem to contribute to groundwater microbial diversity and community composition, although we observed very little to no contribution of surface recharge microbial communities to groundwater communities. The same was observed between groundwater and surface discharge communities. This is very likely due to the severe environmental changes between both habitats. Further temporal studies containing more time points during the year will help confirm our conclusions.

## Figures and Tables

**Figure 1 genes-14-00001-f001:**
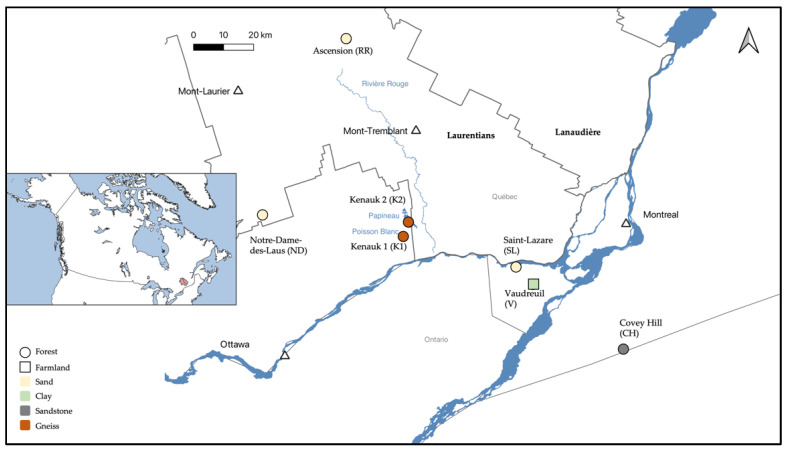
Location of the seven studied aquifers, and the connected surface discharge areas, in the Laurentians and the South of Quebec (Canada).

**Figure 2 genes-14-00001-f002:**
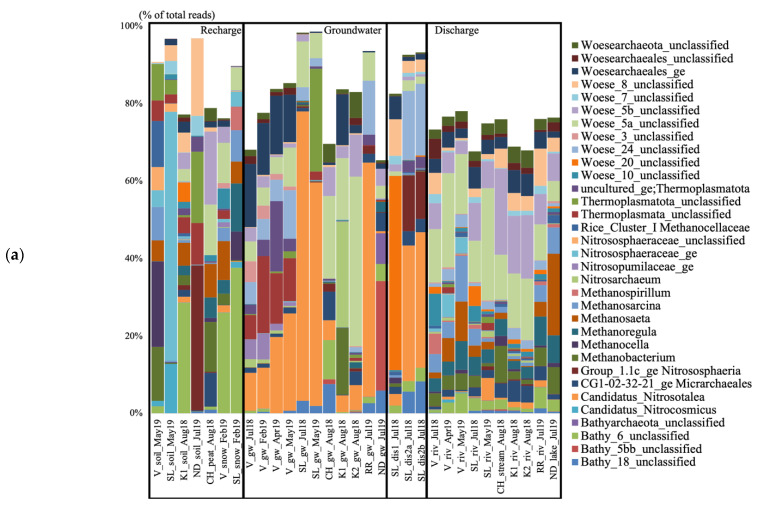
Archaeal (**a**) and bacterial (**b**) taxa based on the 16S rRNA gene diversity at the genus level in the seven studied aquifer systems in Quebec (Canada), represented in percent of the total number of reads. Only taxa representing more than 5% of the total number of sequences are shown. All other taxa are presented colored in white. V, Vaudreuil; SL, Saint-Lazare; CH, Covey Hill; K1, Kenauk 1; K2, Kenauk 2; RR, Rivière Rouge; ND, Notre-Dame-des-Laus; gw, groundwater; dis, discharge; riv, river.

**Figure 3 genes-14-00001-f003:**
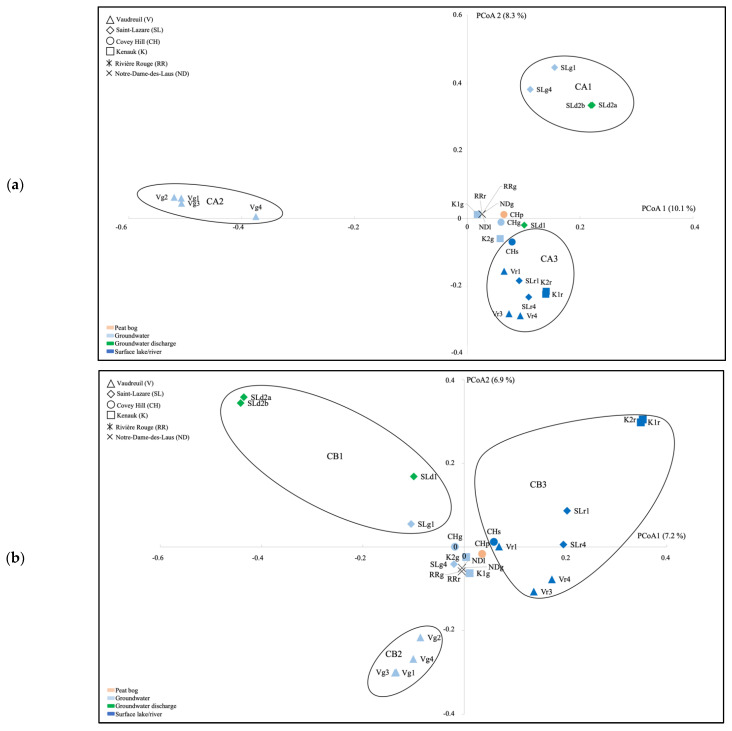
Ordination of the archaeal (**a**) and bacterial (**b**) communities as indicated by principal coordinate analysis (PCoA), based on a Bray–Curtis dissimilarity matrix. V, Vaudreuil; SL, Saint-Lazare; CH, Covey Hill; K1, Kenauk 1; K2, Kenauk 2; RR, Rivière Rouge; ND, Notre-Dame-des-Laus; p, peat bog; g, groundwater; d; groundwater discharge; s, stream; l, lake; r, river; 1, July 2018; 2, February 2019; 3, April 2019; 4, May 2019.

**Figure 4 genes-14-00001-f004:**
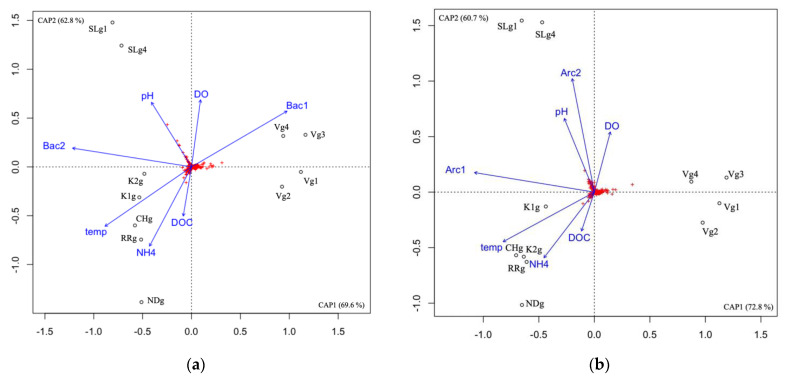
Correlation between the community composition and explanatory factors in the groundwater samples using db-RDA for the archaeal community (**a**), and the bacterial community (**b**). DOC, dissolved organic carbon; DO, dissolved oxygen; temp, temperature; Arc1, archaeal community composition represented by the first axis of a PCoA; Arc2, archaeal community composition represented by the second axis of a PCoA; Bac1, bacterial community composition represented by the first axis of a PCoA; Bac2, bacterial community composition represented by the second axis of a PCoA; V, Vaudreuil; SL, Saint-Lazare; CH, Covey Hill; K1, Kenauk 1; K2, Kenauk 2; RR, Rivière Rouge; ND, Notre-Dame-des-Laus; g, groundwater; g1, groundwater sampled in July 2018; g2, groundwater sampled in February 2019; g3, groundwater sampled in April 2019; g4, groundwater sampled in May 2019. Red crosses represent the ASVs.

**Figure 5 genes-14-00001-f005:**
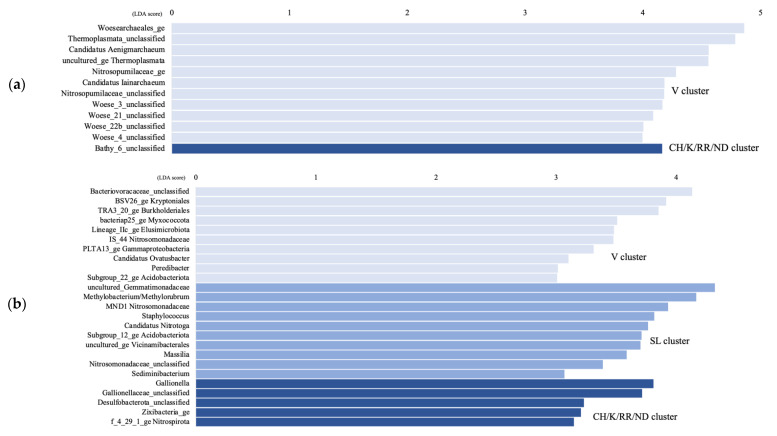
Linear discriminant analysis (LDA) score comparing significantly different archaeal (**a**) and bacterial (**b**) genera between the Vaudreuil (V) groundwater samples, the Saint-Lazare (SL) samples, and the Covey Hill/Kenauk/Rivière Rouge/Notre-Dame-des-Laus (CH/K/RR/ND) samples; calculated by using the LEfSe analysis. Genera with a LDA score > 3.5 are displayed.

**Figure 6 genes-14-00001-f006:**
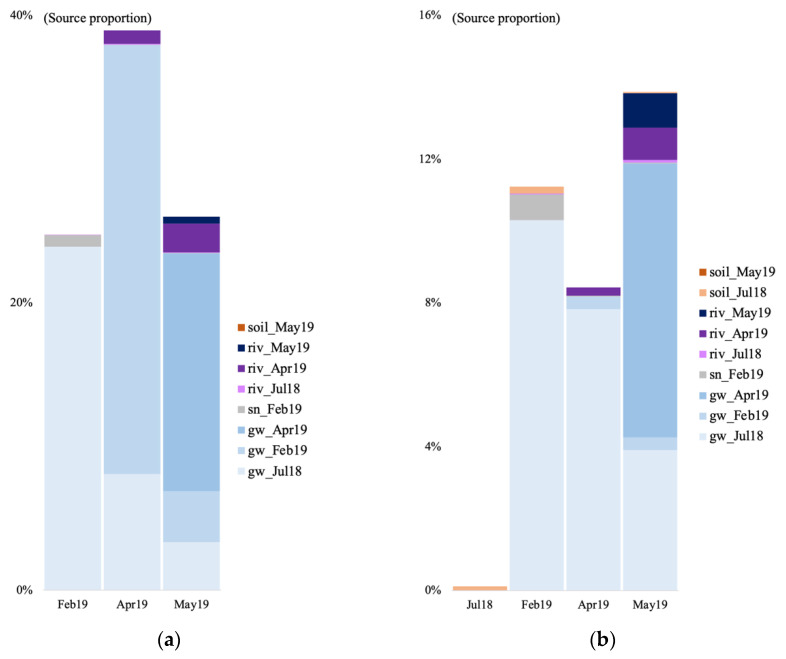
Archaeal (**a**) and bacterial (**b**) community source tracking for groundwater samples in July 2018 (18 July), February, April, and March 2019 (19 February, 19 April, 19 May) at the Vaudreuil site, using fast expectation-maximization microbial source tracking analysis (FEAST). gw, groundwater; sn, snow; riv, river.

**Figure 7 genes-14-00001-f007:**
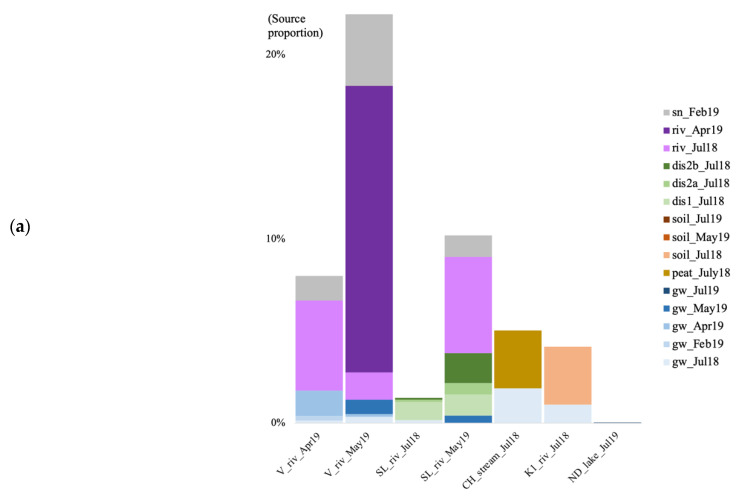
Archaeal (**a**) and bacterial (**b**) community source tracking for surface water samples in July 2018 (Jul18), February, April, May, and July 2019 (19 February, 19 April, 19 May, 19 July) at the Vaudreuil (V), Saint-Lazare (SL), Covey Hill (CH), Kenauk (K), Rivière Rouge (RR) and Notre-Dame-des-Laus (ND) sites, using fast expectation-maximization microbial source tracking analysis (FEAST). gw, groundwater; dis, groundwater discharge; sn, snow; riv, river.

**Table 1 genes-14-00001-t001:** Surface recharge, groundwater and surface discharge samples that were collected, with the associated dates. V, Vaudreuil; SL, Saint-Lazare; CH, Covey Hill; K1, Kenauk 1; K2, Kenauk 2; RR, Rivière Rouge; ND, Notre-Dame-des-Laus; Jul, July; Aug, August; Feb, February; Apr, April.

Site	Surface Recharge	Groundwater	Surface Discharge
V	Soil (Jul 2018) Soil (May 2018) Snow (Feb 2019)	Jul 2018 Feb 2019 Apr 2019 May 2019	River (Jul 2018) River (Apr 2019) River (May 2019)
SL	Soil (July 2018) Soil (May 2018) Snow (Feb 2019)	Jul 2018 May 2019	Forest dis1 (Jul 2018) Forest dis2a (Jul 2018) Forest dis2b (Jul 2018) River (Jul 2018) River (May 2019)
CH	Peat (Aug 2018)	Aug 2018	Stream (Aug 2018)
K1	Soil (Aug 2018)	Aug 2018	River (Aug 2018)
K2	Soil (Aug 2018)	Aug 2018	River (Aug 2018)
RR	Soil (Jul 2019)	Jul 2019	River (Jul 2019)
ND	Soil (Jul 2019)	Jul 2019	Lake (Jul 2019)

**Table 2 genes-14-00001-t002:** Variation in archaeal and bacterial community composition explained by habitat type (groundwater, groundwater discharge, and river surface waters), or site (Vaudreuil and Saint-Lazare), tested by using PERMANOVA.

**Archaea**					
	**Df**	**SumOfSqs**	**R2**	**F**	**Pr(>F)**
Habitat	2	1.7638	0.29862	2.6004	0.001
Site	1	0.7514	0.12721	2.2155	0.002
Residual	10	3.3914	0.57417		
Total	13	5.9066	1.00000		
**Bacteria**					
	**Df**	**SumOfSqs**	**R2**	**F**	**Pr(>F)**
Habitat	2	1.4322	0.23366	1.7254	0.001
Site	1	0.5467	0.08919	1.3172	0.049
Residual	10	4.1505	0.67714		
Total	13	6.1294	1.00000		

## Data Availability

The obtained sequences were deposited in the National Center for Biotechnology Information (NCBI) under the BioProject ID: PRJNA883426.

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
