# Peer review of "From Recharge, to Groundwater, to Discharge Areas in Aquifer Systems in Quebec (Canada): Shaping of Microbial Diversity and Community Structure by Environmental Factors"

_genes, 2022, doi:10.3390/genes14010001_

Round 1

Reviewer 1 Report

The authors wanted to investigate the connectivity between different surface, ground water and discharge microbial (bacterial and archaeal) communities at different geographic locations. They collected data from different seasons and different land use (anthropogenic effect).

Major comment, 

  • Please advance figure 1 and table 1 with a scheme of sample number, land use and other measured abiotic factors (like soil type etc …).

  • I belive figure 3 is over exaggerated since figure 2 explained only 6.41% of the variance.  

  • I urge to reduce the figure numbers, especially figures 4, 8, replace figure 5 with the total bacterial and acheal communities and highlight temporal changes with different genera colors, change figure 6 a,b to one figure. Modify figure 7 legend and make symbols larger 

  • Lines 166-170 please include more description about library size, PE,SE, cleaning etc.

  • Lines 172-179 please provide sequences statistics (number of sequences before after curation etc)

  • Line 180 please provide code in supplementary 

Results

FIgure S3 add y axis description on the figure 

Please add % of variance to the ordinations axis

Line 520-523 please include seasonal variance in the analysis (nested design)

Line 531 please provide rarefaction curve of all the sequences 

General comment

  • I rather separating the result and discussion section, it was hard for me to follow the results

  • The author mentioned land use on different site and it was not presented in the results/discussion or conclusion

Author Response

I rather separating the result and discussion section, it was hard for me to follow the results

Response from authors: We thank the reviewer for this suggestion. This has been done, and now both parts are separate. We believe this improves the flow in the text.

The author mentioned land use on different site and it was not presented in the results/discussion or conclusion

Response from authors: We would argue that land use was added as a variable throughout the discussion, perhaps this was drowned in the text because results and discussion were merged. Please see here for some example, L500-502, L530-533, L605-607.

Please advance figure 1 and table 1 with a scheme of sample number, land use and other measured abiotic factors (like soil type etc …).

Response from authors: We have modified Figure 1, by adding the following information: surface land use (farm or forest), and geological material.

Lines 166-170 please include more description about library size, PE,SE, cleaning etc.

Response from authors: More information was added L190-193.

Lines 172-179 please provide sequences statistics (number of sequences before after curation etc)

Response from authors: This information can now be found in the Table S1.

Line 180 please provide code in supplementary

Response from authors: We added some information for the code used in R, in the Supplementary Material.

Line 520-523 please include seasonal variance in the analysis (nested design)

Response from authors: This is not possible since seasonal variation was only analyzed for one site.

Line 531 please provide rarefaction curve of all the sequences 

Response from authors: In this part of the text, we meant that this sample contained less than 1000 sequences after processing, so it had not been kept for multivariate analyses. The details can now be found in the Table S1.

I belive figure 3 is over exaggerated since figure 2 explained only 6.41% of the variance.

Response from authors: In response to this and the following comments on the number of figures, the previously number figure 3 has been moved to the supplementary material. We agree the number of figures was too high.

I urge to reduce the figure numbers, especially figures 4, 8, replace figure 5 with the total bacterial and acheal communities and highlight temporal changes with different genera colors, change figure 6 a,b to one figure. Modify figure 7 legend and make symbols larger 

Response from authors: As mentioned in response to the previous comment, we have moved most of these figures to the supplementary material. The Figure 7 which is now Figure 3 was also modified to increase symbol size.

FIgure S3 add y axis description on the figure

Response from authors: This was added. It is now Figures S1-S2.

Please add % of variance to the ordinations axis

Response from authors: This information was added in Figure 4.

Reviewer 2 Report

Villeneuve et al manuscript tempt to describe the relationship between the microbial diversity and structure of the aquifer and their recharge and discharge system as well as the impact of seasonal variability. 16S gene sequencing was used to analyze the microbial community of 7 aquifer sites.  Two of them were sampled and studied at different seasons. Physicochemical parameters were analyzed as well. 

The manuscript is hard to read because of the description of each figure which is not necessary. E.g. line 436-444, 537-548. Some of the results are not clearly found in tables and figures, e.g. 6.41% explained archaeal variance from Fig. 3a or 8.64% from Fig. 3b (L 232 and 277, respectively).

The incorporation of a discussion section may improve the manuscript. For E.g. the relevance of the preying observed by the authors could be a separate subtitle in the discussion.

Specific comments:

Shannon index was determined and a graph was included in the supplementary material. The graph does not have an axis title. The results were not commented on in the text. 

L 73. Is it fine to say soil recharge?

L 84-89. The V aquifer was sampled in four opportunities (Table 1) not only in July 2018.

Is D1 in Table 1 the dis1 in Fig. S3? 

L 172. Mothur, uppercase

Fig. 2. What are the red symbols?

L314-317 Please add references to support your assessments.

L341-342. The pH does not decrease in all the samples (Table S1).

L 418. "The river was low..." Improve the writing.

Fig. 6. a) and b) here are associated with bacteria and archaea contrary to the rest of the figures. The numbers in  L425 do seem not to match the graphs. 

Please improve the colour of the figures for a better interpretation.

Author Response

The manuscript is hard to read because of the description of each figure which is not necessary. E.g. line 436-444, 537-548. Some of the results are not clearly found in tables and figures, e.g. 6.41% explained archaeal variance from Fig. 3a or 8.64% from Fig. 3b (L 232 and 277, respectively).

The incorporation of a discussion section may improve the manuscript. For E.g. the relevance of the preying observed by the authors could be a separate subtitle in the discussion.

Response from authors: We thank the reviewer for these comments and the help for improving the text. Also based on similar comments from the other reviewer, we have separated the Results and Discussions part (which were previously merged). We have also reduced the description of the microbial source tracking results L414-481.

Specific comments:

L 73. Is it fine to say soil recharge?

Response from authors: The word was deleted from the sentence.

L 84-89. The V aquifer was sampled in four opportunities (Table 1) not only in July 2018.

Response from authors: A sentence was added L78.

L 172. Mothur, uppercase

Response from authors: We would argue P. Schloss writes it with a lower case, please see his papers from 2009 and 2020 (reference 21).

L314-317 Please add references to support your assessments.

Response from authors: This was added L550.

L341-342. The pH does not decrease in all the samples (Table S1).

Response from authors: Thanks for the good catch. This is a typo, the pH decreased from February to April, as indicated later in the text. We modified this L575.

L 418. "The river was low..." Improve the writing.

Response from authors: The sentence was modified L632-635.

Fig. 2. What are the red symbols?

Response from authors: the red symbols are the ASVs.

Is D1 in Table 1 the dis1 in Fig. S3? 

Response from authors: Indeed, we modified this in Table 1.

Shannon index was determined, and a graph was included in the supplementary material. The graph does not have an axis title. The results were not commented on in the text. 

Response from authors: The axis title was added to what are now Figures S1-S2. The results are commented in the new Results part, L324-333.

Fig. 6. a) and b) here are associated with bacteria and archaea contrary to the rest of the figures. The numbers in L425 do seem not to match the graphs. 

Response from authors: This was done to follow citation in the text, as we present results first for the bacteria. We thank the reviewer for catching that, the numbers were reversed. We corrected this L425-426.

Please improve the colour of the figures for a better interpretation.

Response from authors: Colors of what are now Figures S3-S4 have been modified, as well as what is now Figure 3.

Round 2

Reviewer 2 Report

The manuscript was improved but there are still issues to read the figures and to follow the authors' observations and conclusions.

Several of the graphs do not include axis titles required to understand them. In addition, figures were not cited in the discussion section to help the reader follow the authors´ rationale and conclusions. 

There are some repetitions between the results and the discussion section like L452-458. 

Is there some reference addressing the consolidated and unconsolidated characteristics of each specific aquifer? How do you know the type of each one?

Figure 4. Please explain that the red spots are the ASVs.

Fig. 6 and 7. Please maintain the order of fig a) for archaea and b) for bacteria like in the previous figures.

Author Response

Several of the graphs do not include axis titles required to understand them. In addition, figures were not cited in the discussion section to help the reader follow the authors´ rationale and conclusions.

Reply from the authors: Information was added in the discussion L460, L466, L488, L501, L511, L609, L658. In addition, information was already included L648 and L676. Figures 2, 3a, 4, 5, 6, ad 7 were modified.

There are some repetitions between the results and the discussion section like L452-458.

Reply from the authors: Given the fact that both reviewers suggested separating the results and discussion parts, we would argue here that some repetition of the results is necessary to not force the reader to go back to the results part while reading the discussion.

Is there some reference addressing the consolidated and unconsolidated characteristics of each specific aquifer? How do you know the type of each one?

Reply from the authors: This information was given to us by the hydrogeologists who drilled the wells we used to access the groundwater, and who continue to work on these sites on their own projects. We added this information L85-86 and L98-99.

Figure 4. Please explain that the red spots are the ASVs.

Reply from the authors: The information was added to the Figure 4 legend.

Fig. 6 and 7. Please maintain the order of fig a) for archaea and b) for bacteria like in the previous figures.

Reply from the authors: These changes were made.